# Intelligent Sensor Software for Robust and Energy-Sustainable Decision-Making in Welding of Steel Reinforcement for Concrete

**DOI:** 10.3390/s25010028

**Published:** 2024-12-24

**Authors:** Javier Ferreiro-Cabello, Francisco Javier Martinez-de-Pison, Esteban Fraile-Garcia, Alpha Pernia-Espinoza, Jose Divasón

**Affiliations:** 1SCoDIP Group, Department of Mechanical Engineering, University of La Rioja, 26006 Logroño, Spain; javier.ferreiro@unirioja.es (J.F.-C.); esteban.fraile@unirioja.es (E.F.-G.); 2SCoTIC, Scientific Computation & Technological Innovation Center, University of La Rioja, 26006 Logroño, Spain; alpha.pernia@unirioja.es; 3PSYCOTRIP Group, Department of Mathematics and Computation, University of La Rioja, 26006 Logroño, Spain; jose.divason@unirioja.es

**Keywords:** energy-sustainable decision, resistance spot welding, intelligent welding, machine learning, artificial intelligence (AI)

## Abstract

In today’s industrial landscape, optimizing energy consumption, reducing production times, and maintaining quality standards are critical challenges, particularly in energy-intensive processes like resistance spot welding (RSW). This study introduces an intelligent decision support system designed to optimize the RSW process for steel reinforcement bars. By creating robust machine learning models trained on limited datasets, the system generates interactive heat maps that provide real-time guidance to production engineers or intelligent systems, enabling dynamic adaptation to changing conditions and external factors such as fluctuating energy costs. These heat maps serve as a flexible and intuitive tool for identifying robust operational points that balance quality, energy efficiency, and productivity. The proposed methodology advances decision-making in welding processes by combining robust predictive modeling with innovative visualization techniques, offering a versatile solution for multiobjective optimization in real-world industrial applications.

## 1. Introduction

Welding is a fundamental process in manufacturing [1], critical for constructing durable and high-strength structures. With the advent of Industry 4.0, many manufacturing companies have successfully implemented automated systems for monitoring and controlling welding processes, effectively closing the loop from creation to quality control [2,3]. These advancements have leveraged historical data from a variety of processes to develop artificial intelligence (AI) techniques aimed at enhancing productivity and minimizing production costs. Within this context, machine learning (ML) has become an essential tool, enabling the creation of intelligent models that can predict outcomes, define setpoints, plan, control, and even identify defects in real time. Despite the growing use of AI in Industry 4.0, sustainability concerns—particularly energy efficiency and environmental impact—remain a secondary consideration [4]. Consequently, many companies still rely on manual or semi-automated welding, highlighting the need to fully automate these processes to achieve both sustainability and productivity gains.

Among the various welding techniques available, resistance spot welding (RSW) is particularly economical and productive for joining steel in reinforced concrete structures, as it forms a strong and durable hidden joint within a short processing time [5]. RSW is widely preferred for high-production-rate applications, but it poses technical challenges due to the need for precise control of parameters like current, pressure, and time. Additionally, the high energy demands of RSW increase operational costs and impact environmental sustainability, making the optimization of energy consumption while maintaining quality and productivity a key priority for manufacturers [6].

In this study, we focus on assembly welding for reinforced steel used in concrete structures, which is critical for ensuring structural integrity and durability. This RSW process involves applying pressure and electrical current to overlapping steel bars, forming a solid-state bond. Typical configurations require welding bars of different diameters to ensure joint integrity. The equipment used in this study, PRAXAIR MPH Digital Pneumatic, allows precise parameter control, but optimization is critical to balance quality, cost, and sustainability. Two key indicators, deformation limitation and strength, were selected to evaluate the process. Deformation limitation helps mitigate damage from residual stresses, while strength ensures overall reliability. These metrics are essential for meeting structural standards.

The aim of this work is to develop an intelligent decision support system to optimize the RSW process for steel reinforcement welding. This system integrates machine learning models trained on experimental data to generate intelligent heat maps. These heat maps provide real-time guidance, enabling operators to identify robust operational points that satisfy constraints related to quality, energy consumption, and production time. The main contributions of this study are twofold:The development of machine learning models to estimate optimal welding parameters despite the small dataset available [7]. The proposed machine learning approach accurately predicts ideal parameters and remains robust under varying conditions.The introduction of intelligent heat maps as a visual and flexible tool that enables production engineers to address the challenge of balancing welding quality, production time, and energy savings while adapting to dynamic constraints and external factors like energy costs. Unlike traditional optimization methods, heat maps allow engineers to dynamically identify optimal operating points and robust regions where process parameters remain stable. These maps can also be integrated into intelligent systems for automated decision-making, providing a versatile solution for multiobjective optimization in welding and similar industrial processes.

In Section 2, Section 2.1 presents related works on RSW. Section 2.2 outlines the main issues encountered when working with small datasets, as well as background on ML techniques relevant to this research. Section 2.3 provides a detailed description of the dataset underlying this study, including the experimental procedures and methodologies employed in its collection and development. Section 2.4 explains the proposed approach. Section 3 presents the experimental results and a discussion of these findings. Finally, Section 4 concludes this paper by summarizing the main contributions and suggesting future research directions.

## 2. Materials and Methods

### 2.1. The Resistance Spot Welding Process

Artificial intelligence (AI) has significantly advanced the understanding and optimization of the resistance spot welding (RSW) process [8,9]. Techniques such as neural networks [3,10] and adaptive neuro-fuzzy inference systems (ANFISs) [9] have been employed to predict weld strength in steel sheets with high accuracy. Additionally, carbon emissions from welding processes have been modeled to assess environmental impacts [11].

The recent surge in manufacturing data has facilitated the application of machine learning (ML) in welding quality prediction. For instance, logistic regression models have been utilized to predict defects in electric resistance welded tubes, yielding satisfactory results and enabling the estimation of safe operating ranges for process variables [12]. Support vector regression has also been applied to predict quality in electron beam welding [13]. Furthermore, feature engineering-based ML approaches have been proposed for quality prediction in RSW of steel sheets, combining engineering and data science perspectives to provide insights into welding data and account for the dynamic nature of RSW due to factors like cooling time and electrode wear [14].

In recent years, there has been a growing interest in integrating AI with sensor data to enhance RSW processes. For example, a study developed a machine learning tool to predict the effect of electrode wear on weld quality, achieving an accuracy of 90% [15]. Another research utilized a convolutional neural network (CNN) combined with a long short-term memory (LSTM) network and an attention mechanism to detect welding quality online, achieving a detection accuracy of 98.5% [16]. Additionally, AI-driven interpretation of ultrasonic data has been explored for real-time quality assessment in RSW, highlighting the potential of adaptive welding using ultrasonic process monitoring backed by AI-based data interpretation [17].

Despite these advancements, the application of AI in the RSW of reinforcement bars (rebar) remains underexplored. Moreover, existing studies primarily focus on predictive modeling and do not incorporate visualization techniques to facilitate sustainable decision-making beyond the outputs of ML models. This gap underscores the need for research that integrates AI-driven predictive models with intuitive visualization tools to support informed and sustainable decisions in RSW processes.

### 2.2. Machine Learning-Based Strategy for Small Datasets

Automating complex industrial processes, such as resistance spot welding (RSW), often encounters the challenge of limited data availability during initial implementation phases. In many cases, comprehensive process data are scarce, necessitating the acquisition of information through meticulously designed experiments that span the entire operational state space. Typically, these datasets consist of process conditions as inputs and corresponding weld quality parameters as outputs. Machine learning (ML) models can be developed from this data to predict optimal setpoints based on predefined requirements. However, constructing and deploying these models is particularly challenging when relying on experimental datasets, primarily due to their limited size and scope.

The high economic and time costs associated with experimental data collection often result in datasets that are insufficiently large to capture the full spectrum of process variability. This limitation can lead to the development of overfitted models—models that have learned the training data too well and, consequently, fail to generalize to new, unseen conditions [18]. Overfitting is a common issue in models with high complexity, such as deep neural networks with numerous parameters or extensive regression trees, where the model effectively ’memorizes’ the training data rather than learning the underlying patterns.

Conversely, employing less flexible algorithms, such as simple linear regression, may yield models with inadequate accuracy, rendering them unsuitable for capturing the complexities of the process. Therefore, selecting an appropriate algorithm and meticulously tuning its parameters are critical steps in developing accurate ML models, especially when dealing with small datasets [19]. Robustness is a key consideration to ensure that the setpoints derived from these models remain reliable and stable, even in the presence of input perturbations.

Recent studies have explored various strategies to address the challenges associated with small datasets in ML applications. Techniques such as data augmentation, transfer learning, and the use of ensemble methods have been investigated to enhance model performance and generalization capabilities [20]. Additionally, the integration of domain knowledge into the modeling process has been shown to improve the reliability of predictions, particularly in industrial applications where expert insights can compensate for limited data [21].

In summary, effectively managing the constraints of small datasets requires a balanced approach that combines careful algorithm selection, parameter tuning, and the incorporation of domain expertise. Such strategies are essential to develop robust and accurate ML models capable of supporting automation in industrial processes like RSW.

#### 2.2.1. The Strategy in This Study

The primary task in developing a machine learning (ML)-based decision support system involves testing various models and optimizing their hyperparameters. When working with large datasets and numerous hyperparameters, it is common practice to limit the number of models tested and use grid search for hyperparameter optimization due to the substantial computational cost of each experiment.

In this study, however, the dataset is relatively small (298 rows with five input variables and two target variables). This smaller dataset size has three key implications:There is an elevated risk of overfitting.The limited dataset size necessitates additional experimental trials.Feature selection is unnecessary, as there are only five input variables.

To mitigate the risk of overfitting in training, we employ a nested cross-validation approach. Given the small dataset size, this method is repeated multiple times to enhance the robustness of the results. Additionally, hyperparameter optimization is performed using Bayesian optimization rather than the more conventional grid search, which improves efficiency and allows for more refined parameter tuning.

Beyond traditional models like random forest (RF) and multilayer perceptron (MLP), we also evaluate a powerful AutoML tool, Autogluon, to further streamline and enhance the modeling process.

#### 2.2.2. Bayesian Optimization

Bayesian optimization [22] is a popular algorithm for hyperparameter optimization in machine learning models. While alternative methods like evolutionary algorithms [23] and particle swarm optimization [24] exist, they are typically more computationally expensive and often require numerous evaluations.

In Bayesian optimization, a probabilistic model of the objective function is iteratively updated to guide the selection of hyperparameters. This process involves evaluating the model with selected hyperparameters, updating the probabilistic model based on performance, and repeating until an optimal set of hyperparameters is identified. By leveraging past iterations, Bayesian optimization focuses the search on promising regions of the hyperparameter space.

Compared with standard methods like grid and random search, Bayesian optimization provides a more efficient exploration of the hyperparameter space. Unlike grid search, which examines a fixed set of points and may miss key regions, Bayesian optimization uses its probabilistic model to adaptively explore high-potential areas, even in the presence of non-convex or noisy objective functions. This adaptability helps it avoid local optima and manage evaluation noise more effectively. However, since Bayesian optimization relies on iterative learning from previous evaluations, it tends to have higher computational costs per iteration.

#### 2.2.3. Nested Cross-Validation

Cross-validation (CV) is essential when developing machine learning models, as it provides a reliable estimate of model performance on unseen data and helps mitigate overfitting. The core idea of cross-validation is to split the dataset into training and testing sets multiple times. The model is trained on each training set and evaluated on the corresponding testing set. Repeating this process across different data splits yields a more robust estimate of the model’s performance. The most common method, *k*-fold cross-validation, divides the dataset into *k* equally sized folds. The model is trained on k−1 folds and tested on the remaining fold. This process is repeated *k* times, so each fold is used once for testing.

The development of AI/ML systems is prone to errors, especially related to *model biases* and *dataset shifts* [25]. Model biases often stem from improper training procedures, leading to skewed inference results. Dataset shifts, on the other hand, arise from mismatches between training and testing data distributions. These issues are particularly challenging with small datasets, where the limited data volume amplifies the risk of overfitting and makes mistakes more likely.

Nested cross-validation is a technique designed to evaluate both the generalization error of a machine learning model and its hyperparameters. It is an extension of the *k*-fold cross-validation approach and is particularly beneficial when working with small datasets and models with numerous hyperparameters to tune [26]. Algorithm 1 presents a pseudo-code version of the process.
**Algorithm 1** Pseudo-code for nested cross-validation  1:Split the data into *k* outer folds.  2:**for** each outer fold **do**  3:      Split the outer training set into k−1 inner folds and one inner validation fold.  4:      **for** each combination of hyperparameters **do**  5:            Train the model on the k−1 inner folds.  6:            Evaluate the model on the inner validation fold.  7:       **end for**  8:       Select the hyperparameters that yielded the best performance on the inner validation folds.  9:       Retrain the model on the full outer training set (using all inner folds) with the selected hyperparameters.10:       Evaluate the model on the outer test fold.11:**end for**12:**return** Average performance of the model across all outer test folds.

The key concept in nested cross-validation is the use of two *nested loops* of cross-validation:Outer loop: The outer loop splits the data into training and testing sets, similar to standard cross-validation. However, rather than using the same hyperparameters across all folds, a unique set of hyperparameters is selected for each fold.Inner loop: The inner loop is dedicated to hyperparameter selection. It performs cross-validation on the training set to assess the model’s performance with various hyperparameter configurations. The set of hyperparameters that performs best in the inner loop is then used to train the model on the full training set in the outer loop.

Considering the scalability challenges of nested cross-validation with large datasets, it is noted that for fully automated systems capable of generating extensive data, standard cross-validation may provide a more computationally efficient alternative while maintaining robust performance metrics, a point explicitly addressed in this study.

### 2.3. The RSW Process Dataset

This study utilizes a dataset on the strength of resistance spot welding (RSW), created through controlled welding experiments and described in detail by Ferreiro-Cabello et al. [7]. The following lines summarize the process of creating the dataset and expand on its main features, incorporating additional context and methodology details.

The dataset includes welding tests on rebar of structural steel B500-S, specifically in diameters commonly used in reinforced concrete structures: 8, 10, 12, and 16 mm.

This study focuses on assembly welding, specifically the fabrication of reinforcement steel (ferralla), which does not have a structural purpose.

These diameters were selected to represent a range of joint thicknesses between 16 mm and 32 mm, as shown in Table 1. For each combination, the test configuration consisted of welding three bars of diameter *B* to one of diameter *A*, as illustrated in Figure 1. This configuration ensured that the dataset would include the most frequently used joint dimensions in the production of electro-welded meshes.

The welding experiments were conducted using a PRAXAIR MPH Digital Pneumatic welder, equipped with a maximum power output of 50 kVA. This equipment enabled precise adjustments of key welding parameters, including time (*t*), current intensity (*I*), and pressure (*p*). Welding time was controlled by a digital system, which allowed for cycles of up to 2 s, split into different active time percentages to match the joint thickness. Three discrete values of pressure (5, 6, and 7 bars) were used, and current intensity varied from 40% to 80% of the maximum power, as shown in Figure 2.

A total of 2160 welds were performed, corresponding to 360 combinations of *t*-*I*-*p* for each joint thickness. For each parameter combination, two sets of six welded bars were prepared to assess both strength and deformation characteristics. The welding parameter commands are shown in Table 2.

To evaluate joint deformation, the contact area thickness was measured after welding. This measurement was conducted using a Palmer micrometer to ensure non-destructive sampling accuracy. The deformation (*D*) was calculated as
(1)$$D=1-\frac{\frac{\sum_{n=1}^{6}e_{n}}{6}}{A+B}\%$$
where $e_{n}$ represents the individual thickness measurements. An acceptable weld was defined by minimum values of S=3500 N and D=0.85(A+B), filtering out any outliers due to test errors.

In addition to thickness measurements, weld strength was tested using destructive shear (τ) and tensile (σ) tests with a Servosis ME-420/20 machine. For each combination, the mean values of shear ($\bar{\tau }$) and tensile ($\bar{\sigma }$) strength were calculated. The combined strength (*S*) was computed using
(2)$$S=\sqrt{{\bar{\sigma }}^{2}+{\bar{\tau }}^{2}}$$
as illustrated in Figure 1. This comprehensive testing provided reliable data on joint performance across various conditions.

Finally, the dataset consists of five input variables (diameters *A* and *B*, *t*, *I*, and *p*) and two output variables (*S* and *D*). Extreme outliers were removed, resulting in a final dataset of 298 rows, each representing a unique welding condition.

Figure 3 shows the distribution of joint thickness across experiments, confirming the dataset’s balanced coverage for different joint thicknesses. This balance ensures that the dataset represents a diverse range of welding conditions applicable to industrial RSW processes.

### 2.4. Methodology

The proposed methodology involves a series of steps designed to optimize the machine learning (ML) models, ensure their robustness, and facilitate sustainable decision-making. The steps are as follows:**Selection of ML algorithms**: Choosing the most appropriate algorithms based on the nature of the problem and the limited dataset size.**Model training and evaluation**: Implementing nested cross-validation combined with Bayesian optimization for hyperparameter tuning to enhance model performance and prevent overfitting.**Robustness analysis**: Assessing model robustness to ensure reliable performance across various data conditions.**Heat map generation**: Developing interactive heat maps to support sustainable decision-making by visualizing optimal operational regions.

#### 2.4.1. Selection of the ML Algorithms

The main goal was to develop machine learning (ML) models capable of making robust and accurate predictions of *S* (strength) in Newtons (N) and *D* (deformation) as a percentage (%). As described in Section 2.3, the dataset consists of 298 experimental cases with five input variables representing each test configuration: *A*-*B*-*t*-*I*-*p*, along with two output target variables: *S* (strength) and *D* (deformation).

To enhance model training, the distribution of the target variables was improved by applying a square root transformation. This adjustment aimed to reduce skewness and improve the stability of the predictions. Finally, the entire dataset was normalized to ensure uniformity in the input scale across all variables.

The following algorithms were selected for experimentation due to their suitability for small datasets: ridge regression (Ridge), kernel ridge regression (KRidge), decision tree regressor (DTR), random forest regressor (RF), support vector regression (SVR), and multilayer perceptron (MLP). All of these algorithms are available in the scikit-learn Python library and are well suited for the dataset size and structure. Table 3 summarizes the hyperparameter optimization ranges for each model.

In addition to these established ML models, the AutoGluon AutoML tool was also included in the experiments to explore potential performance gains through automated model selection and tuning. Other algorithms, such as Lasso regression, K-Nearest Neighbors regression, and Gaussian process regression, were excluded from the final experiments due to suboptimal performance with this dataset.

#### 2.4.2. Training and Evaluation

Once the models were selected, we proceeded with their training and evaluation. Given the small dataset size, which increases the risk of overfitting but allows for multiple experimental trials, we incorporated two advanced techniques: Bayesian optimization and nested cross-validation. Although widely recognized, these techniques are not typically part of standard AI model development, where simpler approaches such as grid search and basic cross-validation are more common.

Bayesian optimization was particularly suited to this problem due to its balance between performance and computational efficiency, making it ideal for the dataset size and the need for a robust cross-validation scheme.

For model validation, we employed a repeated *k*-fold nested cross-validation to ensure both accurate hyperparameter tuning and reliable performance estimation. Specifically, we performed a 10-times repeated 10-fold nested cross-validation. In each repetition, the outer loop ran 10 times, creating 10 unique test sets. For each iteration of the outer loop (i.e., with a fixed test set), we conducted a 9-fold cross-validation within the inner loop, using Bayesian optimization with 200 iterations to fine-tune the hyperparameters. In the inner loop, the non-test data were divided into 9 parts: 8 for training and 1 for validation. This inner cross-validation process was repeated 9 times, and the best hyperparameters identified were used to retrain the model on the combined training and validation sets before evaluating it on the test set. Predictions on the test set were stored for subsequent analysis. Figure 4 illustrates an example of the inner loop in one iteration of the outer loop. This process resulted in a total of 100 different training, validation, and testing partitions.

Overall, this nested cross-validation process was repeated 10 times, leading to 900 executions of the inner loop for hyperparameter optimization and 100 executions of the outer loop. The final predictive performance was reported as the mean Root Mean Square Error (RMSE) obtained from the 100 outer loop iterations.

#### 2.4.3. Robustness Analysis

Robustness analysis is crucial in machine learning (ML) applications within the industrial sector, as it ensures models are reliable, accurate, and compliant, thereby mitigating risks associated with incorrect predictions. Sensitivity analysis serves as a key tool in assessing the robustness of ML models by examining how variations in input data influence model outputs. By systematically altering input variables and observing the resulting changes in predictions, one can identify scenarios where the model may underperform and evaluate its reliability under diverse conditions.

Several methods are commonly employed for conducting sensitivity analysis in machine learning:**One-at-a-time (OAT) method**: This approach involves altering one input variable at a time while keeping all other inputs constant to observe the effect on the model’s output.**Morris method** [27]: This technique systematically and randomly varies each input variable to generate a set of samples. Statistical methods are then applied to determine the importance of each input variable.**Sobol method** [28]: A more sophisticated approach that utilizes sequences of orthogonal arrays to systematically vary input variables, enabling the identification of significant input variables and their interactions.

In our approach, we leverage predictions obtained from the testing set using the best-performing algorithms for targets *S* (strength) and *D* (deformation). For a given pressure (*P*) and thickness (*t*), along with specific values of *A* and *B*, we present figures displaying the mean predicted values and their 95% confidence intervals for each pair of current (*I*) and time (*t*) values. These intervals are compared with actual observed values; if the actual value falls within the prediction interval, it indicates the model’s robustness for those input parameters. See Figure 5, Figure 6, Figure 7, Figure 8 and Figure 9.

#### 2.4.4. Creation of Heat Maps

Finally, a strategy was developed to utilize the ML models in a way that facilitates multicriteria decision-making, incorporating welding quality, production time, and energy savings. As previously explained, weld quality is defined by its strength *S*, provided it does not exceed an acceptable deformation value *D*. Additionally, another important objective is to reduce the time required for each weld (*t*) to maximize production. Lastly, minimizing energy costs, which are proportional to the product C=It, serves as another key criterion. Depending on the specific production requirements, these priorities may vary; for example, sometimes the primary goal is to ensure high-quality welds, while in other cases, reducing energy usage to cut costs may take precedence. The importance of this last factor has increased substantially due to recent surges in electricity prices.

At first glance, optimizing the resistance spot welding process to balance energy consumption, efficiency, and quality appears to be a multiobjective optimization problem. However, in practice, different circumstances may require prioritizing one objective over the others. This is why we considered heat maps as a valuable tool to aid in selecting the most suitable settings at any given time. Although ML models can be employed for fully automated production, production engineers often need to ensure that the process operates within a stable region that meets predefined standards for time, quality, and energy efficiency. Thus, the proposed decision support system (DSS) utilizes heat maps generated from ML model predictions to identify, based on joint thickness, the optimal operating points according to weld quality criteria, limited by a maximum deformation $D_{max}$ and a minimum strength $S_{min}$.

As discussed in Section 2, relying exclusively on heat maps may sometimes be restrictive, as additional information or multiple visualization techniques are often required. To address this, we propose an enhanced method that overlays additional data onto the heat maps, aiding decision-making without overwhelming the viewer. Specifically, our approach involves generating heat maps for a given thickness, with intensity (*I*) and time (*t*) represented on the axes. The colors on the heat map depict the strength, predicted by the AI models. To further enhance decision robustness, the color values correspond to the lower bound of the 95% confidence interval ($CI95_{min}$) for *S*, and the upper bound ($CI95_{max}$) for *D* is shown as region borders within the heat map. This design allows decision-makers to quickly interpret the data and make informed choices. By combining heat maps with AI model predictions and confidence intervals, this approach provides a comprehensive solution for visualizing data and selecting optimal values for time and intensity, taking into account required strength, confidence, risk of disturbances, and energy costs. An example heat map is shown in Section 3.

The heat map construction follows these steps:Using the previously selected algorithm, the entire dataset is utilized to search for optimal hyperparameters via Bayesian optimization with 10-fold cross-validation.Once the hyperparameters that minimize the cross-validation error are obtained, 3000 iterations are conducted to create a series of training datasets through random sampling with replacement. In each iteration, a test dataset is also generated with multiple random pairs of *I* and *t*, constrained within plausible physical ranges for a specific thickness and pressure. At the end of the 3000 iterations, a sufficiently large prediction dataset is generated, containing multiple predictions for each combination of input features by aggregating all test predictions.With this dataset, and once thickness and *p* are set, heat maps can be created with matrices of $CI95_{min}$ for *S* and $CI95_{max}$ for *D* across the full range of possible values for *I* and *t*.

## 3. Results and Discussion

This section presents the evaluation of the proposed methodology, focusing on its ability to optimize resistance spot welding (RSW) processes for steel reinforcement bars. The main objective is to balance three critical factors—weld quality, production efficiency, and energy consumption—by leveraging machine learning models and heat maps as decision support tools. The approach includes selecting robust predictive models, validating their generalization across different welding conditions, and utilizing interactive visualizations to aid decision-making. The results are discussed in terms of the accuracy, robustness, and practical applicability of the models, as well as the insights provided by the heat maps for multiobjective optimization.

It is worth emphasizing that the dataset used in this study comprised the typical thicknesses commonly employed in reinforcement fabrication. During the dataset creation process, certain thickness combinations failed to meet the minimum quality criteria due to weak or failed joints. Consequently, fewer viable records were available for extreme thickness combinations, which were limited to specific welding conditions. To overcome this limitation, a robust model selection strategy based on nested cross-validation was implemented, significantly improving the generalization capabilities of the machine learning models. The effectiveness of this approach across the entire range of thicknesses is demonstrated in Figure 8 and Figure 9.

### 3.1. Search of the Best Algorithm for *S* and *D*

The first step, as outlined in the methodology, involved selecting the best ML algorithms and conducting the training and validation of the models (Steps 1 and 2 of the methodology).

Table 4 and Table 5 present the error metrics for the best models obtained with each algorithm, ordered from lowest to highest Root Mean Square Error (RMSE). These tables also include the Mean Absolute Error (MAE) and Mean Absolute Percentage Error (MAPE) for each model. The last column details the hyperparameters optimized through Bayesian optimization. Reported values correspond to the mean and standard deviation (in parentheses) of 100 measurements, collected from each iteration of the outer loop in a 10-repeated 10-fold nested cross-validation. Additionally, Figure 5 and Figure 7 show comparative box plots for both target variables.

The selection of the best algorithm was based on a combined analysis of the mean and variance of RMSE. For *S*, the multilayer perceptron (MLP) was chosen, as it demonstrated a favorable balance between accuracy and robustness. Although AutoGluon achieved a slightly lower mean RMSE than MLP (2085.5 vs. 2104.3), the MLP model exhibited a significantly lower standard deviation (363.7 vs. 462.6), and none of the MLP predictions exceeded an RMSE of 3000, as shown in Figure 6. This indicates that MLP models not only achieved high accuracy but were also more robust, with considerably lower variability. Similar patterns were observed for MAE and MAPE, reinforcing the robustness of the MLP model.

For *D*, the MLP obtained the lowest mean RMSE (0.0109) and a standard deviation value of 0.0036, very close to that achieved with RF. MLP also obtained the best MAE and MAPE.

### 3.2. Model Robustness Analysis

After selecting the best models and evaluating their performance, the next step was to analyze their behavior and variability in response to changes in input variables.

Figure 8 and Figure 9 present examples of the combined models’ predictions for *S* (strength) and *D* (deformation), respectively, across different combinations of *I* (intensity) and *t* (time) with p=6 and for six of the seven thickness levels in the dataset. Note that *I* and *t* values are expressed as percentages of the welding machine’s maximum intensity and a maximum welding time of 2 s, respectively. The cases are ordered first by increasing electrical intensity (*I*), and within each intensity level, by increasing *t*. Consequently, cases located further to the right correspond to higher energy levels, as energy is proportional to the product of *I* and *t*. The solid blue line represents the mean prediction for each group of cases defined by *I* and *t*, while the dashed lines show the 95% confidence interval (CI95) of the estimate. The red line indicates the average actual value for each case.

These graphs demonstrate that, for each thickness, an appropriate selection of *I* and *t* enables a balance between achieving adequate weld strength and maintaining quality. For *S*, the lower bound of the CI95 ($CI95_{min}$) is the critical value to examine, as it represents the most conservative prediction for strength. Conversely, for *D*, the upper bound of the CI95 ($CI95_{max}$) is of interest, as it reflects the worst-case scenario for deformation.

The results indicate that the models exhibit robust behavior, as evidenced by narrow confidence intervals, with the red line (actual values) generally falling within these intervals. However, for the 18 mm thickness, particularly at high energy levels (i.e., high *I* and *t* values), the confidence intervals widen, indicating increased uncertainty in predictions for these conditions. Thus, caution is advised when operating with 18 mm thicknesses at elevated energy levels.

Similar patterns were observed for pressure settings of p=5 and p=7.

### 3.3. Heat Maps for Decision-Making

Figure 10 and Figure 11 present two examples of heat maps incorporating *I* (intensity) and *t* (time) to estimate the values of *S* (strength) and *D* (deformation).

To enhance the understanding of these figures, Table 6 provides a detailed description and interpretation of the variables depicted.

To support multicriteria decision-making, the $CI95_{min}$ of *S* and $CI95_{max}$ of *D* are superimposed on these figures. The colored regions represent the $CI95_{min}$ values of *S*, while solid lines indicate the $CI95_{max}$ boundaries for *D*. Specifically, the green line corresponds to $CI95_{max}=0.05$ for *D*, the yellow line to $CI95_{max}=0.10$, and the red line to $CI95_{max}=0.15$.

Figure 10 shows the heat map for thickness=18 and p=6. The area between the green and yellow lines represents a $CI95_{max}$ range of *D* between 0.05 and 0.10. If a maximum *D* of 0.10 is required, position 1 (t=33% and I=64%) would be appropriate, as it ensures a $CI95_{min}$ of *S* within 10,000–12,000N. However, this decision could be reconsidered if the objective is to further reduce time and energy consumption, albeit with a potential reduction in weld quality. For instance, position 2 represents an alternative operating point with shorter times, at the expense of reducing the *S* range to 8000–10,000N.

Point 3 maintains the same *S* range as Point 1 but halves the value of *t*, with a trade-off: higher energy consumption and reduced weld quality, as it falls within the $CI95_{max}$ range of *D* between 0.10 and 0.15. Lastly, Point 4 increases the *S* range to 12,000–14,000N, although it lies within a smaller, isolated region of stability.

Another example is shown in Figure 11 for thickness=26 and p=7. In this case, a viable option is position 1, with t=81% and I=70%, resulting in an *S* range between 20,000 and 22,000 N. Alternatively, other operating points may be chosen to increase *S*: for instance, position 2 achieves a range of 22,000–24,000 N with a reduction in *t* to 50% but an increase in *I* by 10%. Lastly, position 3 extends the time parameter, raising the *S* range to 24,000–26,000 N, though with deformation values between 0.10 and 0.15.

### 3.4. Trade-Offs Between Quality, Efficiency, and Energy for thickness=18 and p=6

Table 7 presents the key parameters for quality, production time, and energy consumption in the welding process with thickness=18 and p=6 (Figure 10). This table highlights critical insights into process performance at different operational points, emphasizing the trade-offs between production efficiency, energy savings, and weld quality. Values highlighted in red indicate the most unfavorable conditions, while those in bold black represent the optimal values.

In the time-related parameters, the column *Inc. Prod* (%) quantifies the production increase relative to the most unfavorable case, which corresponds to the longest welding time (Point 1). For these calculations, it is assumed that 25% of the total production cycle corresponds to welding operations. The percentage increment in production is calculated using the following formula:(3)$$\mathrm{Inc}.\;\mathrm{Prod}(\%)=\frac{t_{\max}-t}{3t_{\max}+t}\times 100\%,$$
where $t_{\max}$ represents the longest welding time and *t* is the welding time for the specific operational point. This metric provides a quantitative comparison of production efficiency gains achieved under different process conditions.

In the energy-related parameters, the column *P* (kVA) is derived based on the welding machine’s maximum power output of 50 kVA, while the column *Reduc. E* (%) represents the percentage reduction in energy consumption relative to the most unfavorable case (Point 4), where energy consumption is at its maximum. This reduction is determined using the following formula:(4)$$\mathrm{Dec}.\;E(\%)=\frac{E_{\max}-E}{E_{\max}}\times 100\%,$$
where $E_{\max}$ is the energy consumption at Point 4, and *E* corresponds to the energy consumption at the respective point. This metric highlights the potential energy savings achievable through optimized process parameters.

Overall, Table 7 provides a comprehensive view of the trade-offs between weld quality, production efficiency, and energy consumption. Depending on production objectives and energy costs at a given time, different points can be selected. For example, Point 4 corresponds to a weld with excellent quality (*S* maximum), although it incurs a high energy cost and achieves only a 3.1% increase in production. In contrast, if quality requirements are not stringent but energy cost reduction is critical, Point 2 can be selected, offering an energy reduction of 31.3%. However, this comes at the expense of having the lowest weld strength (*S*) in the dataset. Point 3, on the other hand, increases production by 12.8% while achieving an energy cost reduction similar to Point 2. If the quality objectives related to *S* and *D* are sufficient, it could represent a balanced choice.

Based on the results obtained, it can be concluded that the selected machine learning models exhibit remarkable robustness under varying input conditions, maintaining stability even when operational parameters fluctuate. This reliability is further enhanced by the confidence intervals visualized in the heat maps, which provide critical guidance for production decisions. These visualizations ensure that decisions are not only optimal but also resilient to disturbances, thereby guaranteeing consistent weld quality and sustained operational efficiency.

Overall, the integration of machine learning models with interactive heat maps effectively addresses the key challenges in optimizing resistance spot welding processes for steel reinforcement bars. This approach enables the identification of stable operational zones tailored to different production priorities, such as energy efficiency, weld quality, or production speed, while also enhancing adaptability to dynamic scenarios. In conclusion, the combination of robust predictive tools with dynamic visualizations highlights the practical value of this methodology, offering a flexible and reliable framework for decision-making in industrial contexts.

## 4. Conclusions

This study introduces an innovative machine learning-based methodology to optimize resistance spot welding (RSW) processes for steel reinforcement bars, addressing critical challenges in balancing weld quality, energy efficiency, and productivity. By leveraging advanced predictive models and interactive visualization tools, the proposed system empowers production engineers to make adaptive and data-driven decisions, even under dynamic industrial conditions.

The novelty of this approach lies in its integration of predictive modeling with interactive heat maps as user-friendly visualization techniques, significantly enhancing its practical value. By enabling users to pinpoint optimal parameters that align with priorities such as quality, efficiency, or processing time, this methodology facilitates adaptive decision-making in diverse industrial contexts.

Future research could explore real-time sensor integration and adaptive control systems to further enhance its applicability in complex environments. Additionally, developing advanced visualization techniques tailored to high-dimensional data could open up new possibilities for intuitive decision-making.

In conclusion, this study highlights the transformative potential of AI-driven decision support systems in industrial automation, presenting a robust and scalable framework for developing reliable machine learning models, even in data-constrained scenarios. By incorporating intuitive graphical tools for parameter optimization, this approach not only addresses key challenges in sustainability and operational efficiency but also paves the way for smarter and more sustainable manufacturing practices.

## Figures and Tables

**Figure 1 sensors-25-00028-f001:**
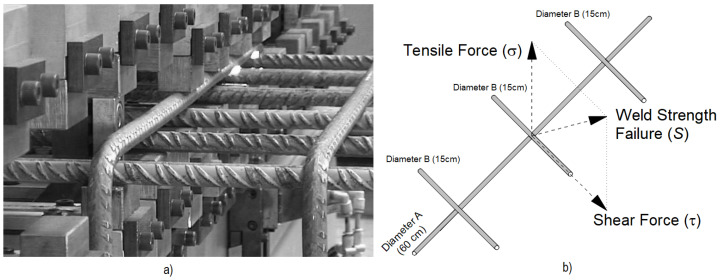
(**a**) Fusion welding of reinforcement bars. (**b**) Test bar configuration and weld strength failure (*S*).

**Figure 2 sensors-25-00028-f002:**
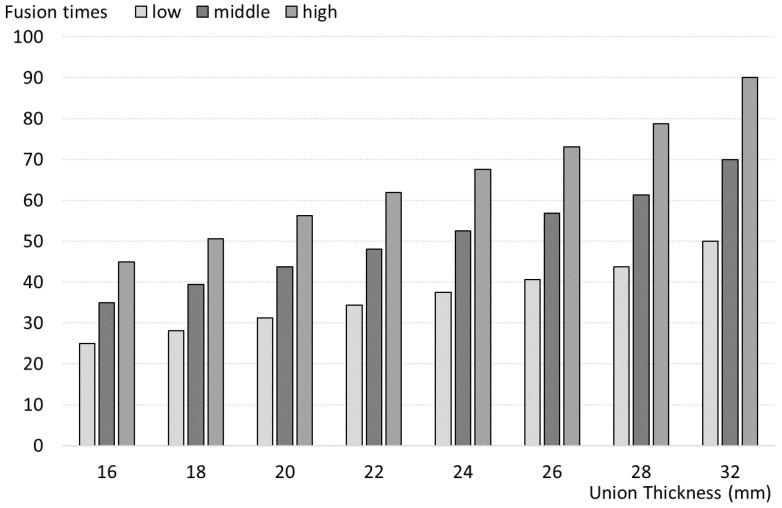
Active welding time according to joint thickness, represented on a 0–100 scale.

**Figure 3 sensors-25-00028-f003:**
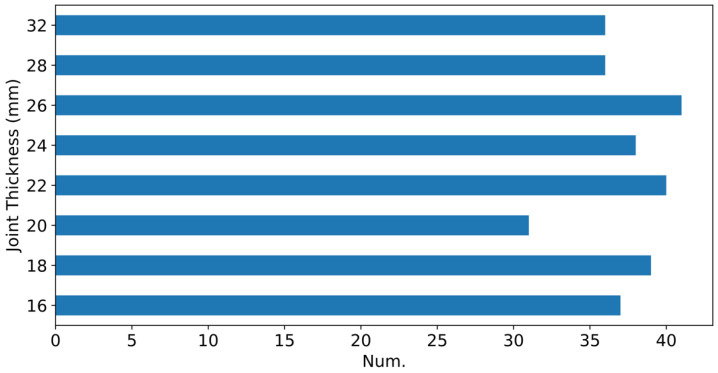
Distribution of experiments according to the total joint thickness, thickness=A+B.

**Figure 4 sensors-25-00028-f004:**
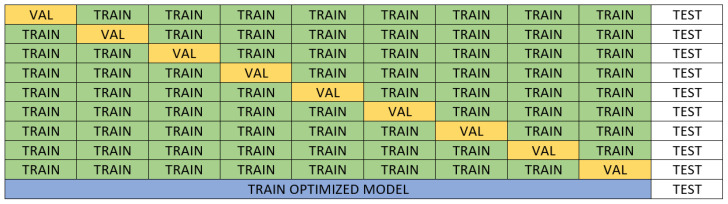
Example of the inner loop within one iteration of the nested cross-validation process. The model is optimized using Bayesian optimization over 9 folds, while the test set remains untouched during this stage. The final model is evaluated on the test fold.

**Figure 5 sensors-25-00028-f005:**
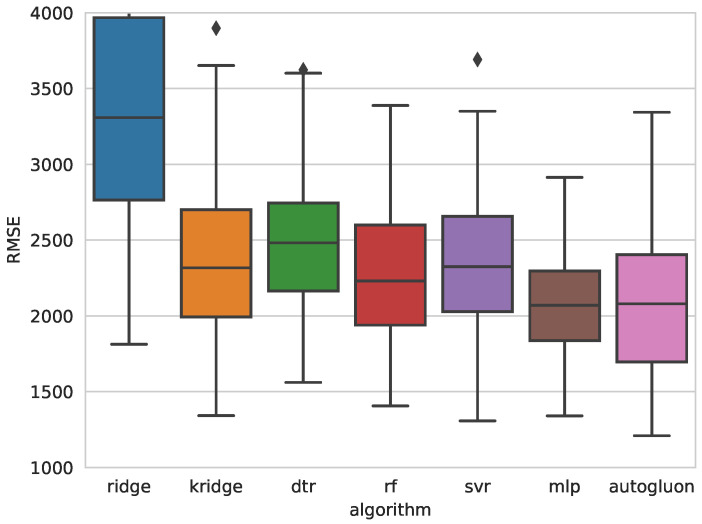
Boxplots of the RMSE obtained with each algorithm for predicting weld strength.

**Figure 6 sensors-25-00028-f006:**
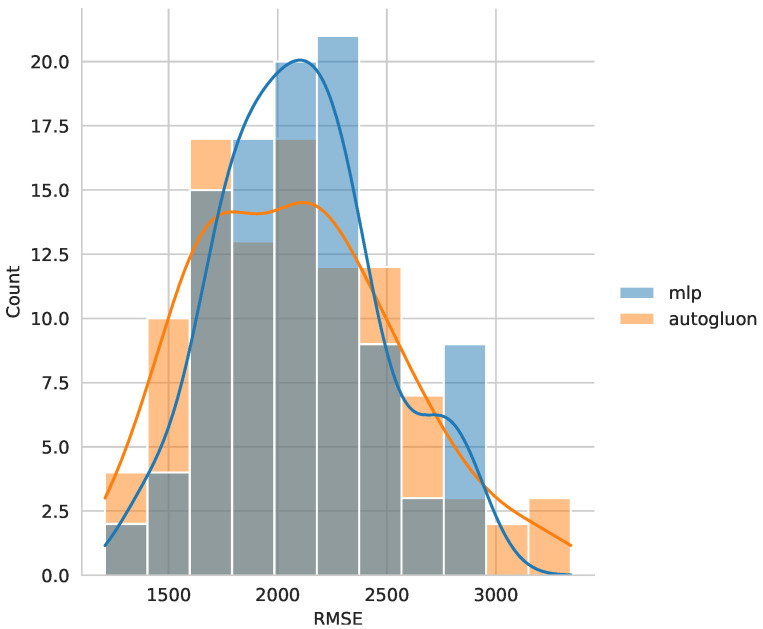
Comparison of the RMSE distribution obtained with the MLP and Autogluon algorithms for weld strength.

**Figure 7 sensors-25-00028-f007:**
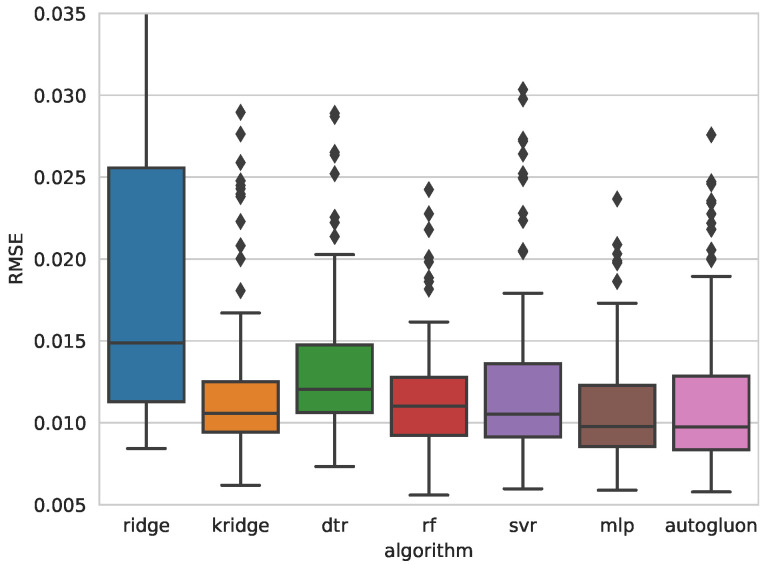
Boxplots of the RMSE obtained with each algorithm for predicting weld deformation.

**Figure 8 sensors-25-00028-f008:**
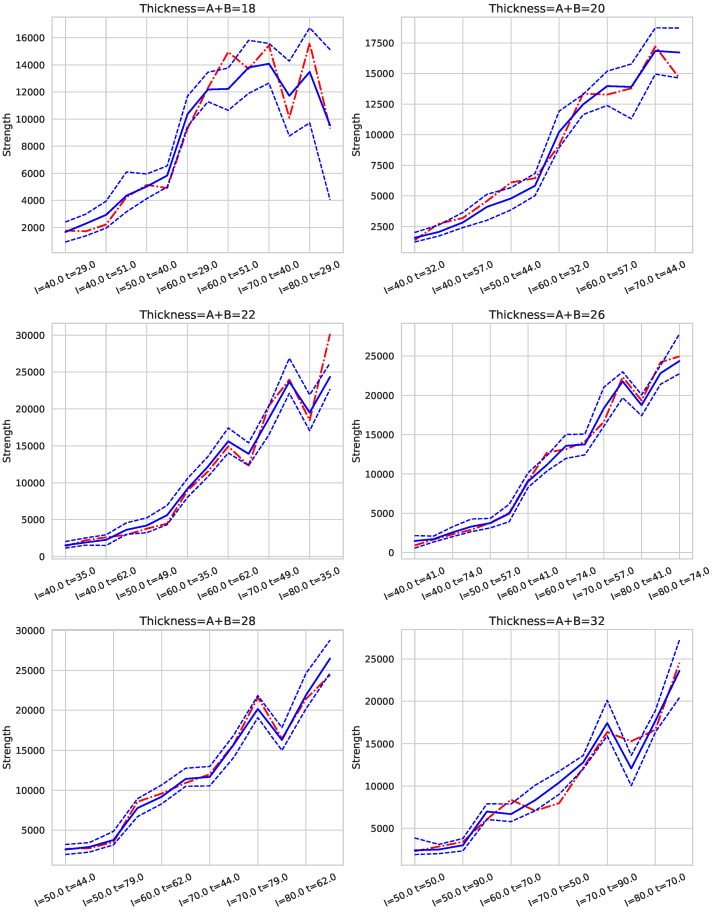
Prediction of *S* with p=6 for each pair (I,t) and six thickness values. The dashed lines indicate a 95% confidence interval and the continuous line is the average value. The red dashed line corresponds to the real value for each pair of values (I,t). The values of *I* and *t* correspond to the % of the maximum intensity and % of the maximum time in 2 s.

**Figure 9 sensors-25-00028-f009:**
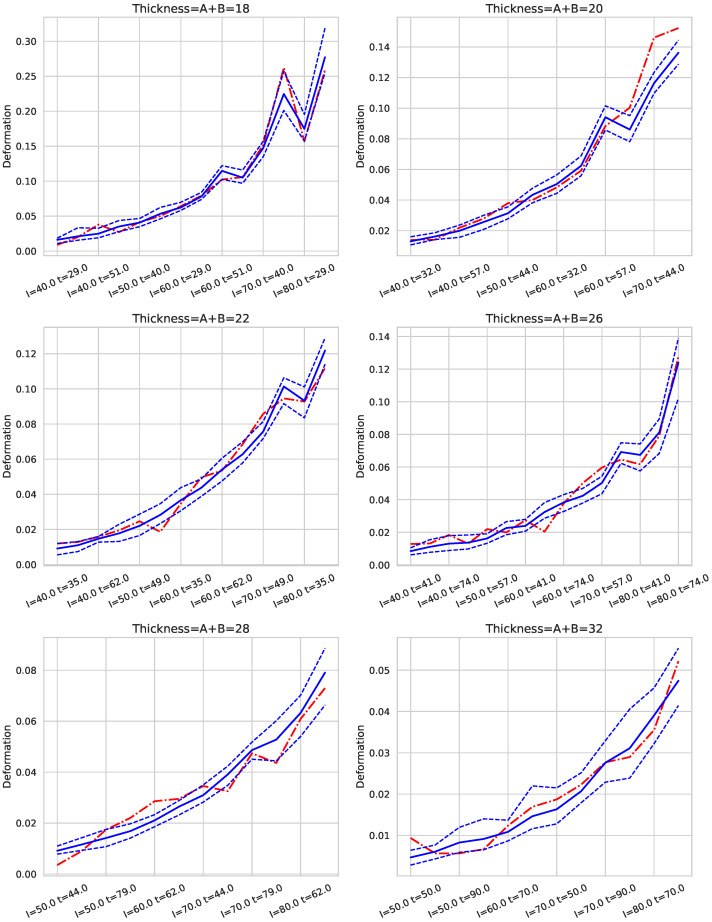
Prediction of *D* with p=6 for each pair (I,t) and six thickness values. The dashed lines indicate a 95% confidence interval and the continuous line is the average value. The red dashed line corresponds to the real value for each pair of values (I,t). The values of *I* and *t* correspond to the % of the maximum intensity and % of the maximum time in 2 s.

**Figure 10 sensors-25-00028-f010:**
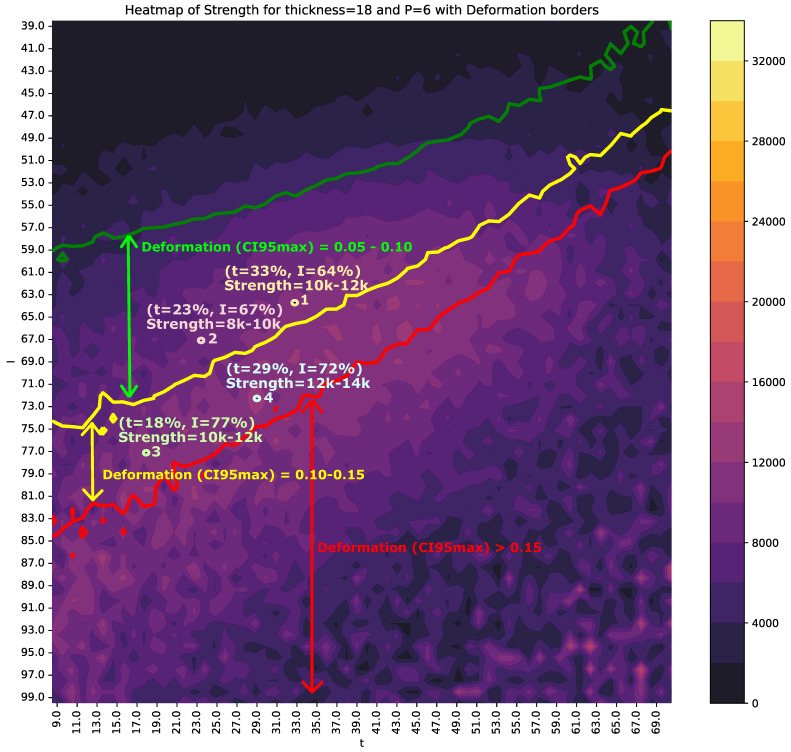
Heat map showing the $CI95_{min}$ value of *S* and $CI95_{max}$ of *D* for each combination of *I* and *t*, with thickness=18 and p=6. The multicolored vertical strip represents the $CI95_{min}$ value of *S*, while the green, yellow, and red contour lines indicate $CI95_{max}$ values of *D* at levels of 0.05, 0.10, and 0.15, respectively. The values of *I* and *t* correspond to the percentage of the maximum intensity and the percentage of the maximum time (2 s), respectively.

**Figure 11 sensors-25-00028-f011:**
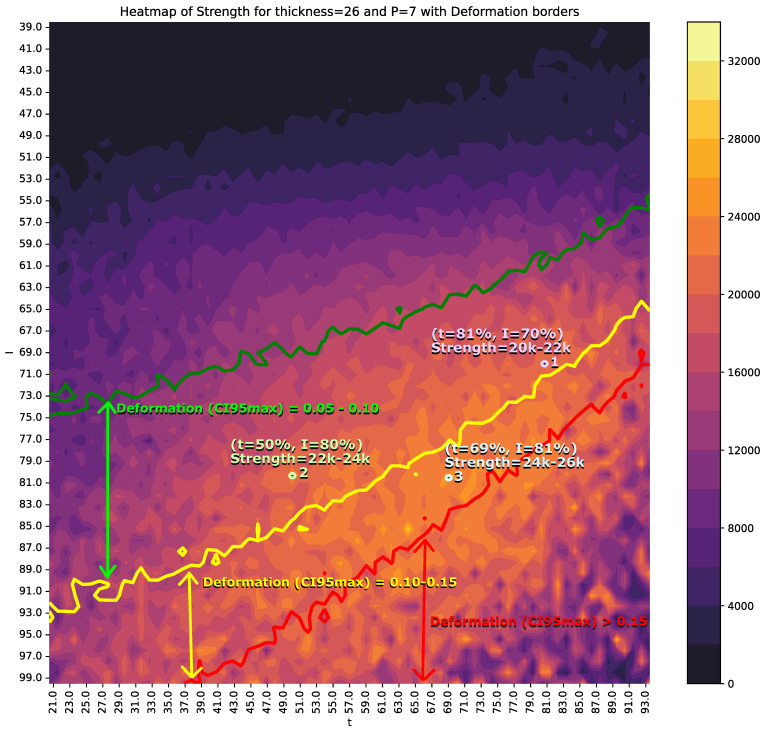
Heat map displaying the $CI95_{min}$ value of *S* and $CI95_{max}$ of *D* for each combination of *I* and *t*, with thickness=26 and p=7. The multicolored vertical strip represents the $CI95_{min}$ value of *S*, while the green, yellow, and red contour lines denote $CI95_{max}$ values of *D* at levels of 0.05, 0.10, and 0.15, respectively. The values of *I* and *t* are expressed as percentages of the maximum intensity and maximum time (2 s) of the welding machine.

**Table 1 sensors-25-00028-t001:** Combinations of diameters for joint Thickness.

Diameter A (mm)	Diameter B (mm)	Joint Thickness (mm)
8	8	16
8	10	18
8	12	20
10	12	22
12	12	24
10	16	26
12	16	28
16	16	32

**Table 2 sensors-25-00028-t002:** The welding parameter commands for the A-t-p equipment.

Joint Thickness (mm)	Equipment Parameters		No. Specimen
	Phase-1 (Contact)	Phase-2 (Welding)	
16	A*0.5	40–50–60–70–80%	45
	t*1/3	25–35–45	
	5_6_7	5_6_7	
18	A*0.5	40–50–60–70–80%	45
	t*1/3	29–40–51	
	5_6_7	5_6_7	
20	A*0.5	40–50–60–70–80%	45
	t*1/3	32–44–57	
	5_6_7	5_6_7	
22	A*0.5	40–50–60–70–80%	45
	t*1/3	35–49–62	
	5_6_7	5_6_7	
24	A*0.5	40–50–60–70–80%	45
	t*1/3	38–53–68	
	5_6_7	5_6_7	
26	A*0.5	40–50–60–70–80%	45
	t*1/3	41–57–74	
	5_6_7	5_6_7	
28	A*0.5	40–50–60–70–80%	45
	t*1/3	44–62–79	
	5_6_7	5_6_7	
32	A*0.5	40–50–60–70–80%	45
	t*1/3	50–70–90	
	5_6_7	5_6_7	

**Table 3 sensors-25-00028-t003:** ML models and ranges of the optimization of each hyperparameter.

Algorithm	Tuning Hyperparameters
Ridge	$alpha=10^{\left[-7,3\right]}$
Kridge	$alpha=10^{\left[-7,0\right]}$
	$gamma=10^{\left[-7,0\right]}$
	kernel = *rbf*
DTR	max_depth∈[1,30]
	min_imp∈[0,1]
RF	max_depth∈[1,30]
	n_estimators∈[1,50]
	min_imp∈[0,1]
SVR	$C=10^{\left[-7,3\right]}$
	$gamma=10^{\left[-7,0\right]}$
	kernel = *rbf*
MLP	hidden_size∈[1,25]
	$alpha=10^{\left[-7,0\right]}$
	activation = *logistic*

**Table 4 sensors-25-00028-t004:** Errors and best hyperparameters for *S*. Values are the average and the standard deviation (between parentheses) obtained with each algorithm and a 10-repeated 10-fold nested cross-validation (100 measures).

Algorithm	RMSE	MAE	MAPE	Hyperparameters
Autogluon	**2085.5** (462.6)	**1486.9** (291.2)	**16.2** (3.2)	presets = best_quality
				time_limit = 5 min
MLP	2104.3 (**363.7**)	1546.0 (**263.6**)	17.5 (**2.7**)	hidden_size = 7.20 (3.99)
				alpha = $10^{-1.673}$ ($10^{0.513}$)
				activation = *logistic*
RF	2275.1 (486.1)	1623.6 (320.1)	17.3 (3.4)	max_depth = 17.97 (6.48)
				n_estimators = 30.82 (11.74)
				min_imp = $2.8\times 10^{-5}$ ($2\times 10^{-4}$)
KRidge	2350.2 (497.0)	1671.5 (311.2)	19.1 (3.5)	alpha = $10^{-1.296}$ ($10^{0.198}$)
				gamma = $10^{-0.623}$ ($10^{0.138}$)
				kernel = *rbf*
SVR	2351.4 (472.6)	1644.9 (299.0)	18.8 (3.4)	*C* = $10^{0.769}$ ($10^{0.362}$)
				gamma = $10^{-0.665}$ ($10^{0.175}$)
				kernel = *rbf*
DTR	2502.6 (510.6)	1840.3 (356.3)	19.7 (3.9)	max_depth = 14.72 (6.90)
				min_imp = $6.8\times 10^{-4}$ ($7.1\times 10^{-4}$)
Ridge	3381.8 (770.4)	2366.8 (440.1)	25.6 (4.9)	alpha = $10^{-6.062}$ ($10^{2.246}$)
				tol = $1.0\times 10^{-4}$

**Table 5 sensors-25-00028-t005:** Errors and best hyperparameters for *D*. Values are the average and the standard deviation (between parentheses) obtained with each algorithm and a 10-repeated 10-fold nested cross-validation (100 measures).

Algorithm	RMSE	MAE	MAPE	Hyperparameters
**MLP**	**0.0109** (0.0036)	**0.0077** (**0.0016**)	**30.91** (10.09)	hidden_size = 4.42 (2.50)
				alpha = $10^{-2.269}$ ($10^{1.266}$)
				activation = *logistic*
**Autogluon**	0.0114 (0.0048)	**0.0077** (0.0018)	32.09 (11.81)	presets = best_quality
				time_limit = 5 min
**RF**	0.0117 (**0.0035**)	0.0083 (**0.0016**)	33.15 (12.60)	max_depth = 18.78 (6.35)
				n_estimators = 32.43 (10.94)
				min_imp = $5.9\times 10^{-6}$ ($4.2\times 10^{-5}$)
**KRidge**	0.0121 (0.0049)	0.0082 (0.0018)	32.25 (11.49)	alpha = $10^{-3.856}$ ($10^{1.172}$)
				gamma = $10^{-2.037}$ ($10^{0.456}$)
				kernel = *rbf*
**SVR**	0.0125 (0.0055)	0.0083 (0.0020)	32.35 (11.28)	*C* = $10^{2.729}$ ($10^{0.378}$)
				gamma = $10^{-1.789}$ ($10^{0.207}$)
				kernel = *rbf*
**DTR**	0.0133 (0.0044)	0.0097 (0.0018)	40.93 (15.02)	max_depth = 17.64 (7.18)
				min_imp = $5.5\times 10^{-4}$ ($8.0\times 10^{-4}$)
**Ridge**	0.0198 (0.0099)	0.0114 (0.0033)	34.81 (**8.37**)	alpha = $10^{0.148}$ ($10^{0.784}$)
				tol = $1.0\times 10^{-4}$

**Table 6 sensors-25-00028-t006:** Description and interpretation of variables included in the heat maps.

Variable	Description	Interpretation in Heat Maps
*t* (Time)	Represents the welding time, expressed as a percentage of the maximum welding time of 2 s.	Displayed along one axis of the heat maps, *t* influences the energy applied during the welding process.
*I* (Intensity)	The electrical current intensity used during welding, expressed as a percentage of the welding machine’s maximum intensity.	Shown along one axis of the heat maps, *I* contributes to the energy applied during the welding process.
*S* (Strength)	Represents the weld strength, calculated as a combination of shear and tensile strength.	The heat maps show the lower bound of the 95% confidence interval ($CI95_{min}$) for *S*, representing the most conservative prediction for weld strength.
*D* (Deformation)	Describes the deformation of the weld, expressed as a percentage of the total thickness of the joint.	The heat maps display the upper bound of the 95% confidence interval ($CI95_{max}$) for *D*, reflecting the worst-case scenario for deformation.
$CI95_{min}$	The lower bound of the 95% confidence interval for *S*.	Indicates the most conservative prediction for weld strength, used to assess the reliability of strength predictions. The multicolored vertical strip represents this variable.
$CI95_{max}$	The upper bound of the 95% confidence interval for *D*.	Indicates the worst-case scenario for weld deformation, guiding decisions to minimize deformation. The green, yellow, and red contour lines denote values of this variable at levels of 0.05, 0.10, and 0.15, respectively.

**Table 7 sensors-25-00028-t007:** Quality, production times, and consumed energy corresponding to the points in Figure 10 of a process with thickness=18 and p=6. The bold values correspond to the best results obtained for each variable.

	Quality		Time			Energy			
Point	*S* (kN)	*D* (%)	*t* (%)	*t* (s)	Inc. Prod (%)	*I* (%)	*P* (kVA)	*E* (kJ)	Reduc. *E* (%)
1	10–12	**5–10**	33	0.66	0.0	64	32.00	675.84	10.1
2	8–10	**5–10**	23	0.46	8.2	67	33.50	516.235	**31.3**
3	10–12	10–15	18	0.36	**12.8**	77	38.50	533.61	29.0
4	**12–14**	10–15	29	0.58	3.1	72	36.00	751.68	0.0

## Data Availability

The datasets presented in this article are not readily available because the data are part of an ongoing study.

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
