# Peer review of "Intelligent Sensor Software for Robust and Energy-Sustainable Decision-Making in Welding of Steel Reinforcement for Concrete"

_sensors, 2024, doi:10.3390/s25010028_

Round 1

Reviewer 1 Report

Comments and Suggestions for Authors

1 In this paper, the evaluation of welding performance from the strength and welding deformation considerations, but the post-weld residual stress will also greatly affect the welding performance indicators, increase the corresponding indicators, more helpful to the judgment of welding performance.

2 Welding predictions obtained from the dataset, the dataset used in this paper has a small range, the diameter of the steel bar used in the 8mm-16mm welding, construction steel bars are mostly bent and overlapped welds, or butt welds, for such cases can be effectively predicted. The distribution of welds of different thicknesses or welding conditions in the dataset may be unbalanced, which will have some impact on the generalization ability of the model.

3 The nested cross-validation approach is more reliable in the small and medium-sized datasets in this paper, how about the computational efficiency or scalability if it is useful for dealing with larger welded datasets?

4 This paper proposes to judge welding quality, production time and energy saving in combination in the ML model, and the prioritization lacks detailed discussion. Is there an objective function for combined judging, or a comprehensive evaluation index.

5 What exactly does CI95min mean in this paper, please give details.

6 The process of heatmap-assisted decision making can be illustrated by a more detailed case study or practice scenario, which verifies in detail the value of the practical engineering applications of the software proposed in this paper.

Author Response

We would like to thank the Reviewers for this exhaustive and helpful revision. We have modified the document according to their comments.

Reviewer 1:

Comments 1:

  1. In this paper, the evaluation of welding performance from the strength and welding deformation considerations, but the post-weld residual stress will also greatly affect the welding performance indicators, increase the corresponding indicators, more helpful to the judgment of welding performance.

Response 1:

Thank you for your observation. In the specific case analyzed in this study, we focused on assembly welding, specifically the fabrication of reinforced steel for reinforced concrete structures. The deformation limitation serves as an appropriate indicator to mitigate damage caused by residual stress. Meanwhile, the strength indicator is the most suitable metric to evaluate the performance of the welding process conducted. We have clarified this point in the revised manuscript to ensure it is clearly understood.

Comments 2:

  1. Welding predictions obtained from the dataset, the dataset used in this paper has a small range, the diameter of the steel bar used in the 8mm-16mm welding, construction steel bars are mostly bent and overlapped welds, or butt welds, for such cases can be effectively predicted. The distribution of welds of different thicknesses or welding conditions in the dataset may be unbalanced, which will have some impact on the generalization ability of the model.

Response 2:

The welding process addressed in this work is assembly welding (reinforcement fabrication), as clarified in the previous response and clarified in the paper. This type of welding does not serve a structural purpose. Therefore, the dataset focuses on the typical thicknesses used for reinforcement fabrication.

However, during dataset creation, some combinations did not meet the minimum quality criteria, as they resulted in weak or failed joints. Consequently, for extreme thickness combinations, fewer viable records were available, and only under specific welding conditions. The use of a robust model selection strategy with nested cross-validation mitigates this limitation by improving the generalization capability of the models. Nonetheless, validating the models across the entire range of thicknesses is essential, as demonstrated in Figures 8 and 9. These points have been clarified in the paper. Thanks

Comments 3:

  1. The nested cross-validation approach is more reliable in the small and medium-sized datasets in this paper, how about the computational efficiency or scalability if it is useful for dealing with larger welded datasets?

Response 3:

As you rightly pointed out, nested cross-validation is the most suitable methodology when working with small to medium-sized datasets. These types of datasets are particularly common when data is obtained experimentally. However, in fully automated systems, where it is possible to generate very large datasets, standard cross-validation may suffice to achieve reliable performance metrics while balancing computational efficiency and scalability. We have included this consideration in the article to highlight its relevance and implications for larger-scale applications

Comments 4:

  1. This paper proposes to judge welding quality, production time and energy saving in combination in the ML model, and the prioritization lacks detailed discussion. Is there an objective function for combined judging, or a comprehensive evaluation index.

Response 4:

This is an insightful question, and addressing it enhances the utility of the proposed heatmaps. Initially, our approach focused on using an objective function and classical optimization methods. However, we identified significant uncertainty when attempting to combine these three objectives—quality, production time, and energy savings—into a single objective function. In practice, the relative importance of these objectives often varies depending on the specific circumstances. While it is possible to include weighting coefficients to prioritize aspects such as production speed or quality, we proposed a more visual and adaptive solution using heatmaps. These allow production engineers to define process operating points dynamically based on real-time constraints and priorities. Moreover, the identification of robust regions within the heatmap, where variations are minimal, adds another layer of practicality to this method. Finally, it is worth noting that these heatmaps can be integrated into intelligent systems capable of making autonomous decisions. In summary, the information generated by these heatmaps, based on robust AI models, presents a highly versatile and effective solution for addressing this type of multi-objective optimization challenge and similar problems in other domains. We have included this in the paper. Thanks.

Comments 5:

  1. What exactly does CI95min mean in this paper, please give details.

Response 5:

 In this paper, CI95min corresponds to the lower bound of the 95% confidence interval (CI95) for the estimate of S, as it represents the critical value to consider. This lower bound is particularly important because it provides the most conservative prediction for strength. Conversely, for D, the upper bound of the CI95 (CI95max) is the value of interest, as it reflects the worst-case scenario for deformation. These aspects are explained in detail in the subsection "Model Robustness Analysis."

To further enhance clarity, we have added “Table 6. Description and interpretation of variables included in the heat maps.” in the subsequent section, "Heat Maps for Decision-Making," which outlines the parameters depicted in the heatmaps. This addition aims to facilitate the interpretation of the heatmap graphics and provide a clearer understanding of the results presented.

Comments 6:

  1. The process of heatmap-assisted decision making can be illustrated by a more detailed case study or practice scenario, which verifies in detail the value of the practical engineering applications of the software proposed in this paper.

Response 6:

We agree that illustrating the process of heatmap-assisted decision-making through a detailed case study or practical scenario is an excellent way to better understand the decision-making process. To address this, we have included Section “3.4 Trade-offs between quality, efficiency, and energy for thickness=18 and p=6”, where we analyze the practical case presented in Figure 10. This section provides a detailed comparison of the differences in quality, production increments, and energy reductions depending on the selected operational point.

Additionally, we have included Tables 6 and 7 to further enhance the reader's understanding of how the heatmaps guide the decision-making process. These tables provide clear insights into the trade-offs between key parameters, illustrating the practical engineering value of the proposed software in real-world scenarios.

We hope this addition effectively demonstrates the utility and application of the heatmap-assisted decision-making process in engineering contexts.

Thank you for your constructive feedback. We appreciate your suggestion and have made the necessary adjustments to address

Reviewer 2 Report

Comments and Suggestions for Authors

Intelligent system for cost-effective and energy-sustainable decision making in the welding of steel reinforcement for concrete (Manuscript ID: sensors-3343224) - original submission

The Authors present and discuss on the developement of an intelligent sensor-software for decision-support that aids in selecting optimal setpoint parameters for the RSW process used in steel reinforcement welding for concrete structures, via machine learning models trained on historical process data to generate intelligent heat maps. The manuscript is not well written and several aspects need revision.

Title: meaningful
Abstract: superfluous (please rephrase and insist on the novelty and importance of your work)
Keywords: meaningful

1. Introduction
* this section is well written, yet the last paragraph must include some more information about the work presented herein and not just a presentatio of this paper's structure (please remove "The paper is organized as ...").

2. Materials and methods - please remove "2. Background" and rename as such (recommended)
* as a general remark, this section is not well written, and the experimental, theoretical and computational methods still need to be discussed in more detail; it is not necessary to present well-known techniques (references will suffice, and please provide more if available); include here sections "3. The RSW process dataset", "4. Machine Learning-based strategy for small datasets", and "5. Methodology" as subsections/subchapters; please provide additional information on methods, operating principles, devices, compounds used, etc.; avoid redundant text.

3. Results and discussion
* please start the section by describing the main aspects of your work: what are you looking for and what is your plan to achieve it (a few sentences will suffice).
* as a general overview/note on this section on "results and discussions": authors should discuss their results in more detail in a more correlated way, provide more references to support the results (if available) and avoid redundant text; refer to previous work and provide further references where needed.
* please present and discuss in more detail in the text all figures and tables;
* a final paragraph of section 3 should be included, to provide the reader with a brief conclusion of your work/manuscript (and further emphasize the novelty of your approach).

4. Conclusion
* This section is also not well written, and needs to be shortened; the authors need to further emphasize on the novelty and importance of their approach.

Author Response

We would like to thank the Reviewers for this exhaustive and helpful revision. We have modified the document according to their comments.

Reviewer 2:

Comments 1:

Title: meaningful
Abstract: superfluous (please rephrase and insist on the novelty and importance of your work)
Keywords: meaningful

Response 1:

Thank you for your insightful observation regarding the abstract. We have revised it to address your concerns by significantly reducing its length and ensuring a sharper focus on the novelty and importance of the work. The updated abstract now emphasizes the contributions of the study, particularly the development of machine learning-based heatmaps as a novel decision-support tool for optimizing resistance spot welding processes under dynamic conditions.

Comments 2:

  1. Introduction
    * this section is well written, yet the last paragraph must include some more information about the work presented herein and not just a presentation of this paper's structure (please remove "The paper is organized as ...").

Response 2:

Thank you for your valuable observation regarding the introduction. We have revised this section to better clarify the contributions of the work presented herein, ensuring that the focus remains on the core aspects of the study. Additionally, we have streamlined the introduction by reducing its length to improve readability and conciseness.

As suggested, we have removed the phrase "The paper is organized as...".

Comments 3:

  1. Materials and methods - please remove "2. Background" and rename as such (recommended)

* as a general remark, this section is not well written, and the experimental, theoretical and computational methods still need to be discussed in more detail; it is not necessary to present well-known techniques (references will suffice, and please provide more if available); include here sections "3. The RSW process dataset", "4. Machine Learning-based strategy for small datasets", and "5. Methodology" as subsections/subchapters; please provide additional information on methods, operating principles, devices, compounds used, etc.; avoid redundant text.

Response 3:

Thank you for your detailed feedback regarding the "Materials and Methods" section. In response, we have renamed the section as suggested and incorporated the other sections as subsections within this chapter for better organization and coherence.

Additionally, we have thoroughly reviewed the entire text and decided to remove subsection 2.3. "AI-Driven Visualization Approaches," as we determined that it is not essential for understanding the methodology. Also, two subsections have been fused: “working with small…” and “Machine learning-based strategy…”. Finally, we have reordered the subsections. These adjustments help to reduce the overall length of the article while maintaining the clarity and focus of the section.

We appreciate your suggestions, which have contributed significantly to enhancing the structure and conciseness of this section.

Comments 4:

  1. Results and discussion
    * please start the section by describing the main aspects of your work: what are you looking for and what is your plan to achieve it (a few sentences will suffice).
    * as a general overview/note on this section on "results and discussions": authors should discuss their results in more detail in a more correlated way, provide more references to support the results (if available) and avoid redundant text; refer to previous work and provide further references where needed.
    * please present and discuss in more detail in the text all figures and tables;
    * a final paragraph of section 3 should be included, to provide the reader with a brief conclusion of your work/manuscript (and further emphasize the novelty of your approach).

Response 4:

We agree that this section required improvement, and we have made substantial enhancements based on your thoughtful suggestions. Following your guidance, we have reorganized and refined the "Results and Discussion" section to ensure clarity, depth, and coherence. Specifically, we have introduced the section with a concise overview of the main objectives of our work and the strategies employed to achieve them, as recommended.

Additionally, we have significantly improved the discussion of all figures and tables, ensuring that their relevance and implications are clearly articulated. To facilitate a better understanding of the results, we have included Table 6 which provides a comprehensive summary of the key findings. Furthermore, we have incorporated a case study (subsection 3.4 with Table 7) to illustrate potential costs, quality outcomes, and production times, as depicted in Figures 8 and 9. This addition highlights the practical implications of our findings and enhances the contextual relevance of our work.

Finally, we have concluded the section with a dedicated paragraph that summarizes the main contributions of our study, emphasizing the novelty and potential impact of our approach. We are confident that these revisions have substantially improved the quality and presentation of this section. Thank you once again for your insightful comments, which have been instrumental in strengthening this part of the manuscript.

Comments 5:

  1. Conclusion
    * This section is also not well written, and needs to be shortened; the authors need to further emphasize on the novelty and importance of their approach.

Response 5:

Thank you for your observation. We have completely rewritten the conclusions, taking into consideration the recommendations provided. The section is now more concise and emphasizes the novelty and importance of our approach more clearly. We believe this revision enhances the clarity and focus of the manuscript.

Thank you for your valuable feedback. We agree with your suggestion to clarify the motivation and main contribution of our work.

Round 2

Reviewer 1 Report

Comments and Suggestions for Authors

The paper has been revised as required and can now be accepted.

Reviewer 2 Report

Comments and Suggestions for Authors

"Intelligent system for cost-effective and energy-sustainable decision making in the welding of steel reinforcement for concrete" - Manuscript ID: sensors-3343224 (Revision 1) 

The Authors have correctly addressed most of the issues raised during the peer-review procedure, therefore the manuscript is now acceptable for publication.